# Antibacterial Peptides Resistance in *Staphylococcus aureus*: Various Mechanisms and the Association with Pathogenicity

**DOI:** 10.3390/genes12101527

**Published:** 2021-09-28

**Authors:** Miki Kawada-Matsuo, Mi Nguyen-Tra Le, Hitoshi Komatsuzawa

**Affiliations:** Department of Bacteriology, Graduate School of Biomedical and Health Sciences, Hiroshima University, Hiroshima 734-8551, Japan; mmatsuo@hiroshima-u.ac.jp (M.K.-M.); lenguyentrami@hiroshima-u.ac.jp (M.N.-T.L.)

**Keywords:** antimicrobial peptides, bacteriocin, two-component system

## Abstract

*Staphylococcus aureus* is a bacterium that mainly colonizes the nasal cavity and skin. To colonize the host, it is necessary for *S. aureus* to resist many antibacterial factors derived from human and commensal bacteria. Among them are the bacteria-derived antimicrobial peptides (AMPs) called bacteriocins. It was reported that some two-component systems (TCSs), which are signal transduction systems specific to bacteria, are involved in the resistance to several bacteriocins in *S. aureus*. However, the TCS-mediated resistance is limited to relatively low concentrations of bacteriocins, while high concentrations of bacteriocins still exhibit antibacterial activity against *S. aureus*. To determine whether we could obtain highly bacteriocin-resistant mutants, we tried to isolate highly nisin A-resistant mutants by exposing the cells to sub-minimum inhibitory concentrations (MICs) of nisin A. Nisin A is one of the bacteriocins produced by *Lactococcus lactis* and is utilized as a food preservative worldwide. Finally, we obtained highly nisin A-resistant mutants with mutations in one TCS, BraRS, and in PmtR, which is involved in the expression of *pmtABCD*. Notably, some highly resistant strains also showed increased pathogenicity. Based on our findings, this review provides up-to-date information on the role of TCSs in the susceptibility to antibacterial peptides. Additionally, the mechanism for high antimicrobial peptides resistance and its association with pathogenicity in *S. aureus* is elucidated.

## 1. Introduction

Commensal bacteria inhabit the parts of the human body that come in contact with the external environment (oral cavity, digestive organs, vagina, anus, skin, etc.). Commensal bacteria compete and cooperate with each other in the environment. In organs where microorganisms are originally resident, the number of bacteria is controlled by the immune system to prevent infectious diseases. However, in the compromised hosts such as elderly people and patients with systemic diseases, the immune activity in these individuals is considered to be weakened. Under such conditions, the proportion of each bacterial species in some sites of the human body is altered, causing dysbiosis; in some cases, infectious diseases occur. Antibiotics (currently called “antibacterial chemotherapeutic agents”) are used to treat bacterial infections. However, depending on the dose and frequency of antibiotic administration, drug-resistant bacteria sometimes emerge.

*Staphylococcus aureus* is known as a commensal bacterium in humans; it generally localizes in the nasal cavity, skin, and intestine. *S. aureus* is a highly adaptable bacterium causing opportunistic infections, such as suppurative diseases, pneumonia, and sepsis [1,2,3]. Additionally, *S. aureus* causes food poisoning because it produces several heat-stable enterotoxins [2]. *S. aureus* is a pathogenic bacterium with a wide variety of virulence factors, and antibiotic resistance is likely to occur with long-term exposure to antibacterial chemotherapeutic agents [4]. According to the 2013 Centers for Disease Control and Prevention (CDC) report, 80,000 people were affected by methicillin-resistant *Staphylococcus aureus* (MRSA) in the USA in that year. In addition, the O’Neill Report [5] estimated that the number of deaths from drug-resistant bacteria would exceed that from cancer in 2050. Among the infectious disease-causing microorganisms listed in this report are various drug-resistant bacteria, such as MRSA, penicillin-resistant *Streptococcus pneumoniae* (PRSP), and carbapenem-resistant enterobacteriaceae (CRE). In response to these reports, counterplans against drug-resistant bacteria, known as antimicrobial resistance (AMR) action plans, are advocated worldwide.

Antibacterial chemotherapeutic agents are generally administered to cure *S. aureus* infections. However, the emergence of MRSA has become a growing challenge of this treatment approach. Additionally, disinfectants are also widely used for the prevention of nosocomial infection. It is reported that the *qac* genes, which encodes a multidrug efflux pump that expels toxic molecules, contributes to the development of resistance to quaternary ammonium compounds such as benzalkonium chloride [6,7]. In the genus Staphylococcus, the *qac* genes are encoded in a plasmid, and six types of Qac efflux pumps are reported. Among the Qac proteins, QacA and QacB are highly conserved among Staphylococcus species, while QacC, QacG, QacH, and QacJ, which belong to the small multidrug resistance (SMR) family, are known to have amino acid sequence diversity among the Staphylococcus species. Therefore, *S. aureus* shows resistance not only to several antibacterial chemotherapeutic agents but also to the other antibacterial agents such as disinfectants. In recent years, antimicrobial peptides (AMPs) have attracted attention as antibacterial chemotherapeutic agents. These AMPs are derived from various living organisms, such as humans, plants, and bacteria [8,9,10,11]. Bacterial AMPs are also called bacteriocins. Some of these antibacterial peptides and bacteriocins were also shown to be effective against MRSA [12,13,14] and have potential applications in the clinic [15,16]. Therefore, these peptides are attracting attention as candidates for next-generation antibacterial chemotherapeutic agents because of their high stability and the establishment of purification methods in recent years. In this review, we provide up-to-date information for understanding the role of the potentially present strains that found by applying high concentrations of antimicrobial chemotherapeutic agents, focusing on the genetic characteristics and high resistance mechanisms of isolated strains. Then, we explain the pathogenicity of isolated endogenous highly nisin A-resistant strains and the underlying mechanism. This information reveals the existence of these endogenous antibiotic-resistant strains, which may be an “outbreak reserve force”, and it is thought that these results will help suppress the potential emergence of highly resistant strains of *S. aureus*.

## 2. Bacteriocins

Bacteriocins are ribosomally synthesized peptides or proteins that exhibit antibacterial activity against bacterial species that are closely related to bacteriocin producers [17,18]. Bacteriocins are mainly classified into classes I and II [19]. Class I bacteriocins (peptides <5 kDa) are called “lantibiotics” and contain a ring bridged by lanthionine and 3-methyllanthionine residues [20], whereas class II bacteriocins comprise unmodified amino acids [20]. Lantibiotics are subdivided into types A and B [21]. Type A lantibiotics bind to lipid II, which is involved in peptidoglycan synthesis, and then inhibit cell wall biosynthesis and disturb the bacterial membrane [17,20], while type B lantibiotics are globular peptides that inhibit cell wall biosynthetic steps such as transglycosylation [22]. Type A lantibiotics are further classified into two subtypes: type A(I), while lactin 481 and nukacin ISK-1 are classified as subtypes of type A(II) [19]. Class II bacteriocins are classified into the following three subclasses: IIa, IIb, and IIc [23].

Nisin A is a bacteriocin produced by L. lactis [24]. Nisin A is a lantibiotic that contains unusual amino acids such as lanthionine, β-methyllanthionine, and dehydrated amino acids [20]. Nisin A binds to lipid II, resulting in membrane disturbance. Recently, it was reported that nisin A is associated with DNA condensation by interfering with chromosome replication or segregation in *S. aureus* [25]. Nisin A has broad-spectrum antimicrobial activity, mainly against gram-positive bacteria [26,27,28,29,30,31]. Due to its broad-spectrum activity, nisin A is widely used as a food additive worldwide for the prevention of food poisoning [26,32,33]. In addition, Alves DCB et al. reported the potential use of nisin combined with oxacillin for methicillin-resistant *S. aureus* [34]. Bacteriocins, including nisin A, were also investigated as potential antibacterial chemotherapeutic agents for clinical application [26,29,35].

## 3. Two-Component Systems and Their Association with AMP Resistance

Recently, two-component systems (TCSs) were reported to be associated with the resistance to several types of antibacterial agents, such as bacitracin, vancomycin, human β defensins (hBDs), LL37, and bacteriocins [36,37,38,39,40,41]. TCSs are predominantly found in prokaryotes. TCSs comprise a sensory histidine kinase (HK) and a cognate response regulator (RR) [42,43]. The sensor is a transmembrane protein that senses changes in the external environment, resulting in autophosphorylation of histidine residues (HKs) in the sensor and transfer the phosphate to aspartate residues of the cognate response regulator (RR) [43,44]. The phosphorylated RR then binds to target DNA elements with strong affinity, activating or repressing the transcription of target genes. Thus, bacteria are able to quickly adapt to the external environment by regulating the expression of the respective genes.

It was revealed that *S.*
*aureus* has 16 sets of TCSs. The function of each TCS is shown in Table 1. A well-studied TCS is the Agr system, which is known to be widely involved in the regulation of virulence factor expression. Agr has a central role in the quorum-sensing system, which senses cell density via autoinducer peptides (AIPs) [45]. Agr is involved in the expression of many factors, including virulence factors mediated by RNAIII, a gene product of *hld* (delta-hemolysin). RNAIII was demonstrated to directly upregulate *hla* (α-haemolysin) expression [45] and downregulate the expression of *spa* (protein A) [46] and the transcription factor rot gene, which is responsible for the repression of toxins [47]. RNAIII binds to the target mRNA directly, resulting in the up- or down-regulation of gene expression. Phenol soluble modulins (PSMs) were demonstrated to be regulated by AgrA directly and have versatile virulence activities such as epithelial colonization, cytotoxic activity, biofilm formation, and antimicrobial activity [48,49,50]. However, the precise mechanism of the expression of other virulence factors mediated by the Agr system is still unknown. SaeRS, one of the TCSs in *S. aureus*, is also known as a global regulator of virulence factors and is important for the regulation of coagulase, α-toxin, β-haemolysin, γ-haemolysin, staphylococcal immunoglobulin-binding protein, nuclease, leucocidin, toxic shock syndrome toxin-1 (TSST-1), epidermis deprivation toxin, etc. SaeRS promotes the expression of coagulase [51,52,53].

TCSs were reported to be involved in controlling the susceptibility to human-derived AMPs. AMPs are innate immune factors and are produced in various tissues and organs, such as the skin, lungs, and intestines [8,9,10,11,54]. The most well-known AMPs are defensins. Defensins are classified into two types: α-defensins from neutrophils and Paneth cells and β-defensins (hBDs) from mainly epithelial cells [8,10,55]. Another major peptide is human cathelicidin (LL37), which is found in various cells, including neutrophils and epithelial cells [10]. The Aps system is related to resistance against human β-defensin-3 (hBD3), LL37, and bacteriocins (nisin A, nukacin ISK-1), which possess a strong positive charge. ApsR regulates *dlt* and *mprF* (*fmtC*) expression, causing an increase in cell surface charge [36,41,56]. Dlt is involved in alanine addition to teichoic acids in cell walls, while MprF is involved in lysine addition to phosphatidylglycerol in cell membranes [57,58]. Alanylation of teichoic acids and lysyl-phosphatidylglycerol contribute a shift to a weak negative charge on the cell surfaces (Figure 1). Since *apsRS* expression is negatively controlled by Agr, this resistance system is mainly observed in the early stage of bacterial growth with low expression of Agr, while in the stationary phase, Agr expression is increased, leading to suppression of the expression of ApsRS. Therefore, the charge on the surface of the bacterial cells is altered during growth. As a result, the susceptibility to antibacterial peptides changes during growth, with low susceptibility observed in the exponential phase and high susceptibility in the stationary phase [37].

VraSR regulates many factors involved in cell wall biosynthesis and is associated with the susceptibility to cell wall synthesis inhibitors such as β-lactams, vancomycin, cycloserine, teicoplanin, and bacitracin [38,39,59]. Upon the addition of cell wall synthesis inhibitors, VraSR is activated, resulting in the upregulation of several cell wall synthesis genes, including the transpeptidase *pbp2* and the transglycosylase *sgtB* [38].

BraRS, which is involved in the acquisition of bacteriocin resistance, was first discovered to be involved in the resistance to bacitracin, one of the bacteriocins produced by *Bacillus subtilis* [60]. The resistance mechanism involves the sensing of low concentrations of bacitracin by the complex of the BraRS TCS and the upstream ABC transporter BraDE. As a result, the regulator BraR promotes the expression of the ABC transporter VraDE, an intrinsic resistance factor for bacitracin [61]. The BraRS-VraDE system is considered to be a TCS system that widely supports the sensing of several bacteriocins because it is also involved in the resistance to nukacin ISK-1 produced by *Staphylococcus warneri* and nisin A produced by *L. lactis* [40,41]. In addition, it was reported that the ABC transporter BraDE, encoded with BraRS regulon, was also associated with nisin resistance by directly interacting with BraS [62].

In conclusion, regarding bacteriocin resistance, it was clarified that BraRS regulates the expression of ABC transporters to promote resistance against several bacteriocins. ApsRS and VraRS also participate in bacteriocin resistance by changing the charge and increasing the expression of cell wall synthesis genes, respectively (Figure 1) [41]. In this way, it is expected that *S. aureus* performs precise TCS-mediated control to survive even in the presence of many bacteriocins produced by some other bacteria colocalized in the bacterial flora.

Three TCSs are known to be involved in bacteriocin resistance: BraRS, ApsRS, and VraRS. BraRS is a TCS that senses various bacteriocins and induces the expression of the ABC transporter *vraDE*. BraAB is required for BraS to sense bacteriocins, and *braAB* expression is also induced via BraRS (left). ApsRS is involved in resistance to positively charged antimicrobial peptides (AMPs). Aps controls the cell surface charge by regulating the expression of *dlt* and *mprF*. Aps is negatively controlled by Agr, the quorum sensing system (middle). VraRS is involved in resistance to cell wall synthesis inhibitors and regulates several genes in the cell wall synthesis system such as *pbp2*, *sgtB*, and *murZ* (right).

Amino acid sequences of ApsRS and BraRS from *S. aureus* are compared with those from the other staphylococci (Figure 2 and Table 2). Although ApsS in *S. aureus* does not have a high similarity with that of the other staphylococci, the response regulator ApsR in *S. aureus* shows relatively high similarity (above 79% identity) with that of other staphylococcal species except for *S. pseudintermedius*. It is speculated that this system, which changes the surface charge, is widely conserved among staphylococci. On the contrary, BraRS in *S. aureus* shows low similarity with that of the other staphylococci. Therefore, BraRS, which senses nisin A, bacitracin, and Nukacin ISK-1, may be specific to *S. aureus*.

## 4. Isolation of Highly Nisin-Resistant Strains with Point Mutations by Exposure to Sub-MICs of Nisin

As explained above, *S. aureus* has several systems for resistance against AMPs, including some bacteriocins. Since these systems are effective in low concentrations of AMPs, AMPs are sometimes utilized at a high concentration for a treatment or a food preservative. Consequently, *S. aureus* cells may have a chance to be exposed to high concentrations of bacteriocins, leading to the emergence of highly resistant strains. To clarify our hypothesis, we tried to investigate whether *S. aureus* can acquire high resistance by exposing the cells to bacteriocins.

In a previous study, high nisin A-resistant strains were isolated by exposing *S. aureus* to sub-MICs of nisin A and designated them *S. aureus* nisin-resistant (SAN) strains [63]. Some SAN strains showed resistance to not only nisin A but also to the other antibacterial agents such as bacitracin and human AMPs. Our findings suggest that the acquisition of bacteriocin resistance may result in cross-resistance to the other antibacterial agents. *S. aureus* can adapt to not only antibacterial chemotherapeutic agents but also preservatives like nisin A.

Some of the SAN strains showed constitutively high expression of *vraDE*, the expression of which is normally induced by nisin A. In contrast, no induction of *vraDE* expression was observed in the other SAN strains. By analyzing the sequences of *vraDE* and *braRS*, we found point mutations in the BraRS region of three strains with high VraDE expression (Figure 3A). These three strains, designated SAN1, SAN8, and SAN87, showed mutations in different genes in the *braRS* region [63].

On the other hand, no mutation in the *braRS-vraDE* region was observed in the high nisin-resistant strain (designated SAN2), in which *vraDE* expression was not induced upon the addition of nisin A. DNA microarray analysis showed high expression of the *pmtRABCD* gene. By determining the DNA sequence of this region, a point mutation was found in the gene encoding PmtR (Figure 3B) [64].

### 4.1. Mechanism Underlying High Nisin Resistance in the BraXRS Mutant

In the SAN1 type strain, only one point mutation was observed between the –35 and –10 boxes in the *braXRS* promoter region (Figure 3A upper). The strains with mutations in the promoter region showed increased promoter activity, resulting in the high expression of *braRS*. Based on these results, we speculated that a high amount of BraRS in the SAN1 strain conferred increased levels of phosphorylated BraR upon the addition of nisin A, which resulted in increased expression of VraDE compared to that in the wild-type strain (Figure 4). One point mutation occurred in *braS* (SAN87) and *braR* (SAN8), resulting in one amino acid substitution. Due to this mutation, SAN8 and SAN87 showed constitutively high VraDE expression, even in the absence of nisin A. Therefore, the unphosphorylated form of the mutant BraR protein and mutated BraS are capable of inducing VraDE expression (Figure 4).

There were only two reports of increased bacteriocin resistance as a result of mutations in the *braXRS* region. The mutation sites in the *braXRS* region are the promoter region of *braXRS* (one report), the *braS* sensor region (two reports), and the *braR* regulator region (one report). All of the mutant strains showed a single nucleotide point mutation. Although it is known that *vraDE* is regulated by BraRS and its expression is induced in a nisin concentration-dependent manner [41,61], this response is only observed under low concentrations of nisin. Mutations in the *braXRS* region of the TCS may ultimately trigger the development of high nisin resistance.

According to the structural analysis of NsrR, which shares homology with BraR, the mutation site observed in SAN8 type is an aspartic acid residue at the 96th position (Asp to Val) (Figure 3A lower), adjacent to a conserved phenylalanine residue that is a switch residue [65]. In general, in RRs, phosphorylation of aspartic acid causes conformational changes in two amino acids of the RR called switch residues, resulting in dimer formation [41,66]. Due to the different properties of aspartic acid (hydrophilic and acidic) and valine (hydrophilic and nonpolar), the dimerization interface region undergoes a structural change, leading to BraR forming a dimer and binding upstream of *vraDE* even in the nonphosphorylated state.

In the SAN87 type strain, a mutation was found at position 130 (asparagine to lysine) of the sensor protein BraS (Figure 3A middle). The sensor protein is composed of three regions: the sensing region (which includes the transmembrane region), the histidine kinase (His KA) region, and the ATPase region. The mutation site in the sensor protein in SAN87 is in the histidine kinase region, next to the sensing region. Previous reports have isolated nisin-resistant *S. aureus* strains with mutations in the histidine kinase and ATPase regions [67]. Since BraS is necessary to activate the BraR regulator, mutations in these regions may result in the activation of the BraR by increased autophosphorylation, which in turn enhances the induction of target factor gene expression.

### 4.2. Mechanism Underlying High Nisin Resistance in the PmtR Mutant

BraRS-VraDE-independent nisin A high resistant *S. aureus* was also isolated (SAN2, SAN233, SAN455, and SAN469) [64]. This mutant has a mutation in *pmtR*, which encodes a transcriptional regulator that controls the expression of the *pmtR and pmtABCDD* (*pmtA-D*) operon (Figure 3B). As a result, these mutants exhibited increased expression of PmtA-D, a transporter responsible for the export of PSMs. High nisin A-resistant mutants were also isolated from not only the MW2 strain but also the MRSA COL strain and TY34 strain. All these mutants had a point mutation in the *pmtR* gene, yielding a mutant *PmtR* with an amino acid replacement of alanine to aspartic acid (SAN2 from MW2) or a truncated PmtR (SAN469 from MW2, SAN233 from COL and SAN455 from TY34) (Figure 3B).

Previous studies have shown that *PmtR* is a negative transcriptional regulator of the *pmtRABCD* (*pmtR-D*) operon (Figure 5) [68]. The EMSA results showed that the mutated *PmtR* from the SAN2 (point mutation type) and the other three SAN (truncation type) strains lost their ability to bind to the DNA region upstream of *pmtR-D*. The mutated *PmtR* lost their ability to bind to the DNA region upstream of *pmtR-D*, resulting in increased expression of *pmtR-D*. This result suggests two possibilities. One possibility is that the mutation site is important for DNA binding. The other is that mutations alter the structure of *PmtR*, resulting in the loss of DNA binding. In conclusion, the increased expression of *PmtA-D* is involved in high nisin A resistance. Since *PmtA-D* is responsible for the secretion of PSMs, it is speculated that *PmtA-D* excretes some antibacterial agents, including PSMs and human-derived AMPs externally.

## 5. Nisin Resistance Affects Virulence

In addition to having high nisin A resistance, the mutants with BraRS mutations were also found to be highly resistant to bacitracin and gallidermin derived from *Bacillus subtilis* and *Streptococcus agalactiae,* respectively. Since the BraRS-VraDE system was originally involved in the acquisition of resistance to low concentration of these bacteriocins, this acquisition of high resistance was considered to be due to the high expression of VraDE.

Unlike the BraRS mutation, the *PmtR* mutation in SAN2 caused increased pathogenicity. The SAN2 strain showed reduced susceptibility to the innate immune factors hBD3 and LL37 and high haemolytic activity [64]. In addition, in a mouse infection model, the SAN2 strain showed a higher survival rate than the wild-type MW2 strain [64].

In the wild-type MW2 strain, the *PmtA-D* proteins form an ABC transporter consisting of two membrane proteins (*PmtA* and *C*) and two ATPases (*PmtB* and *D*) [69]. This transporter is involved in the transport of PSMs and delta-hemolysin from the cytoplasm to the extracellular space [69,70]. PSM is involved in a wide range of pathogenic activities, such as epithelial colonization, biofilm formation, proinflammatory activity, cytolytic activity, and surface diffusion activity, leading to antibacterial effects [48,49,71]. In addition, Pmt transporters were reported to be associated with human-derived AMPs such as hBD3 and LL37 [72]. In the SAN2 strain, mutated *PmtR* loses its ability to bind to the target DNA region, resulting in a high expression of *pmtRABCD*, followed by enhanced *PmtA-D* function [64]. Although there are no clear structural similarities among PSMs, delta-haemolysin, hBD3 and LL37, it is speculated that *PmtA-D* may be involved in the export of these peptides. From these results, it is clear that the increased expression of *pmtA-D* affects not only the high resistance to antibacterial peptides but also the pathogenicity of *S. aureus*.

## 6. Conclusions

In general, bacteria acquire resistance against antibacterial agents via endogenous mutations and exogenous resistance genes. Several TCSs such as ApsRS, VraRS, and BraRS were demonstrated to be associated with the resistance to antibacterial AMPs, including human-derived AMPs, bacteriocins, and antibacterial chemotherapeutic agents. However, these TCS-mediated resistances are effective to low concentrations of AMPs. We and other laboratories demonstrated that *S. aureus* could acquire high nisin A resistance via endogenous mutations upon exposure to nisin A [63,64,67]. Since several AMPs including nisin A are used in food preservatives and are considered to be candidates for clinical use, the development of AMP resistance in *S. aureus* will continue to occur. It is possible that some portion of AMP-resistant strains become highly pathogenic. Therefore, highly pathogenic *S. aureus* strains are potentially present and may represent an “outbreak reserve”. Antibacterial chemotherapeutic agent abuse and secondary infection followed by unpredictable infectious diseases, such as COVID-19, may potentially enhance the occurrence of *S. aureus* infections. In such situations, we have to suppress the emergence of highly pathogenic *S. aureus* strains. Looking ahead, surveillance and research of endogenous resistant mutant strains of *S. aureus* will help suppress the emergence of potentially highly pathogenic strains.

## Figures and Tables

**Figure 1 genes-12-01527-f001:**
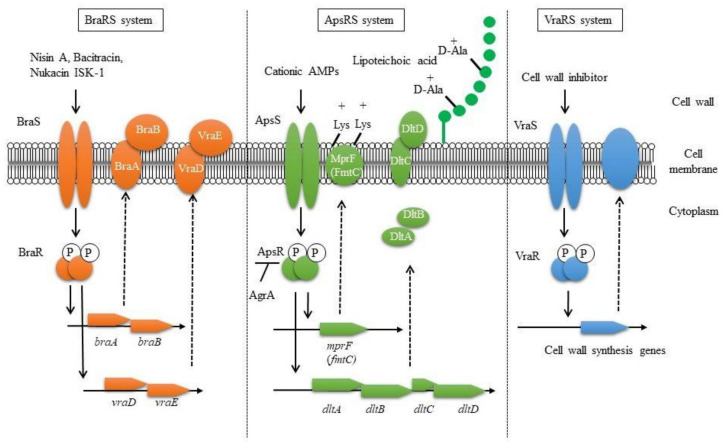
Proposed bacteriocin resistance mechanism mediated by TCSs in *S. aureus*.

**Figure 2 genes-12-01527-f002:**
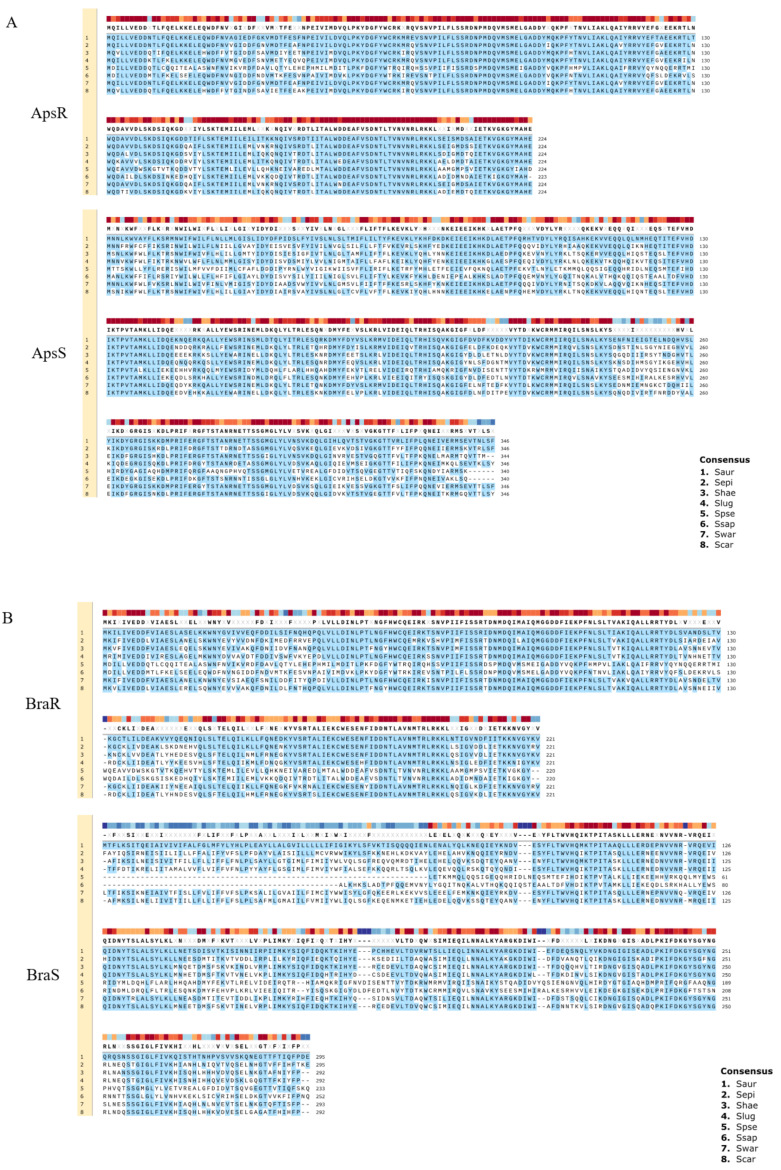
Comparison of two-component systems showing homology with ApsRS and BraRS of *S. aureus* among staphylococci. Alignment of ApsRS (**A**) of BraRS (**B**) among 8 staphylococcal species. 1. *Staphylococcus aureus* (Saur), 2. *Staphylococcus epidermidis* (Sepi), 3. *Staphylococcus haemolyticus* (Shae), 4. *Staphylococcus lugdunensis* (Slug), 5. *Staphylococcus pseudintermedius* (Spse), 6. *Staphylococcus saprophyticus* (Ssap), 7. *Staphylococcus warneri (Swar), 8. Staphylococcus carnosus* (Scar).

**Figure 3 genes-12-01527-f003:**
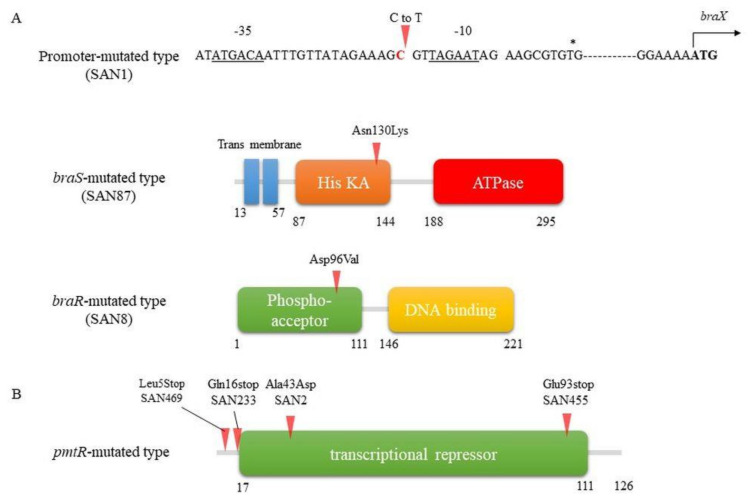
Mutation sites in the high nisin A-resistant strain. (**A**) Mutation sites in the *braXRS* region. The mutation sites in the *braXRS* region in the isolated mutants are indicated by red arrows. The nucleotide sequence upstream of *braXRS* in the MW2 strain is shown, with the –35 and –10 regions indicated by the underline. The transcription initiation start sites are labelled with an asterisk, and the ATG translation initiation codons are indicated in bold. Mutation sites in SAN1, SAN87, and SAN8 are found in the *braXRS* promoter region (upper), His KA of *braS* region (middle), and phospho-acceptor domain of *braR* region (lower), respectively. His KA, dimerisation and phospho-acceptor domain of histidine kinases. (**B**) Mutation sites in the *pmtR*. The *pmtR* in MW2 strain is shown. In the isolated mutants from MW2 (SAN2 and SAN469), COL (SAN233) and TY34 (SAN455) strains, all mutations were found within the *pmtR* (red arrows).

**Figure 4 genes-12-01527-f004:**
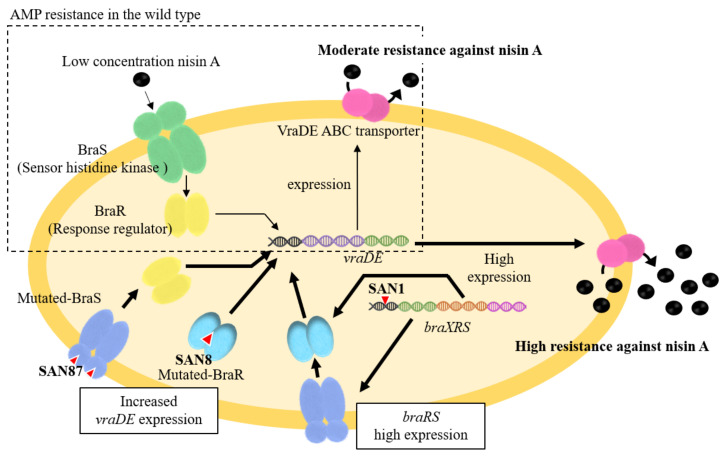
Mechanism underlying the high nisin A resistance of the BraRS mutant. In the MW2 strain, the BraRS-VraDE system responds to low concentrations of nisin A (upper side). In highly nisin A-resistant strains, mutation (red triangle) occurs in the *braXRS* region, resulting in high expression of *vraDE* and resistance to nisin A (lower side).

**Figure 5 genes-12-01527-f005:**
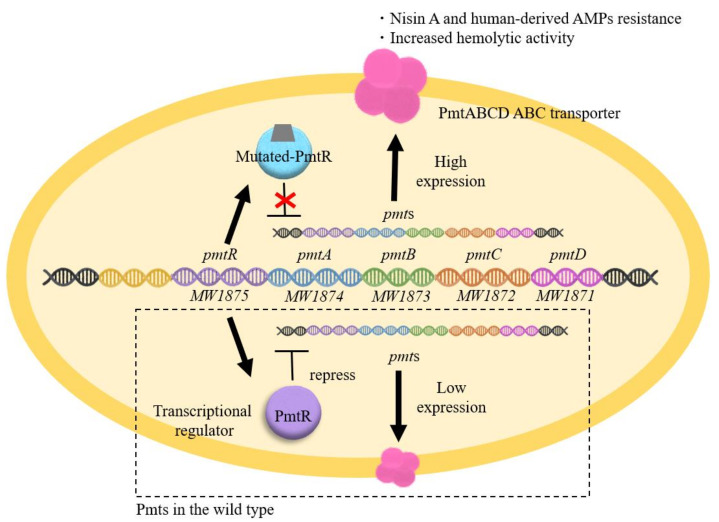
Mechanism underlying the high nisin A resistance of the *PmtR* mutant. In the MW2 strain, *PmtRABCD* expression was low (lower side) because *PmtR* negatively regulates *pmtRABCD* expression. In the *PmtR* mutation strains, mutations in *PmtR* result in high expression of *pmtRABCD* (upper side). As a result, high nisin A resistance and an increase in haemolysis are observed.

**Table 1 genes-12-01527-t001:** Function of two-component system in *Staphylococcus aureus* MW2 strain.

TCS no.	Gene Name	Gene ID	Function
TCS1	*vicRS*	MW0018-19	Cell division/separation, lethal
TCS2	*htpRS*	MW0198-99	response to extracellular phosphates and survival/multiplication within host cells
TCS3	*lytSR*	MW0236-37	Lytic enzymes
TCS4	*apsRS, graRS*	MW0621-22	Bacterial surface charge, dltABCD, *mprF*
TCS5	*saeRS*	MW0667-68	Virulence factor expression
TCS6	*-*	MW1208-09	No report
TCS7	*arlRS*	MW1304-05	Virulence factor expression
TCS8	*srrAB*	MW1445-46	Oxidative stress
TCS9	*phoPR*	MW1636-37	inorganic phosphate uptake
TCS10	*-*	MW1789-90	No report
TCS11	*vraSR*	MW1824-25	Resistant against cell wall synthesis inhibitor
TCS12	*agrCA*	MW1962-63	quorum sensing, virulence factor expression
TCS13	*kdpDE*	MW2002-03	Neutrophil sensitivity
TCS14	*hssRS*	MW2282-83	Iron efflux
TCS15	*nreCB*	MW2313-14	nitrate respiration
TCS16	*bceRS, braRS*	MW2544-45	Bacteriocin resistance

**Table 2 genes-12-01527-t002:** % amino acid sequence identity of ApsRS and BraRS in *S. aureus* compared to eight Staphylococcal species.

*S. aureus MW2*	% Sequence Identity in:						
	*S. epidermidis*	*S. haemolyticus*	*S. lugdunensis*	*S. pseudintermedius*	*S. saprophyticus*	*S. warneri*	*S. carnosus*
ApsR	91.52	85.71	85.71	62.50	79.37	91.52	84.82
ApsS	69.67	67.73	67.05	46.47	60.00	73.70	69.08
BraR	79.64	78.28	76.47	42.53	41.82	81.45	78.28
BraS	60.34	63.01	58.56	30.17	28.68	62.80	61.64

## Data Availability

No new data were created or analyzed in this study. Data sharing is not applicable to this article.

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
