# Peer review of "Antibacterial Peptides Resistance in *Staphylococcus aureus*: Various Mechanisms and the Association with Pathogenicity"

_genes, 2021, doi:10.3390/genes12101527_

Round 1

Reviewer 1 Report

The authors nicely summarize the role of two-component systems in the resistance of S. aureus against various antimicrobial peptides/proteins with focus on their work about Nisin A resistance. They show that the use of sub-MIC of an antimicrobials also lead to the resistance development against other antimicrobial agents, which is an important topic. However, sometimes the review is not clearly written, which makes it difficult to understand their point. Sometimes there should also be more focus on the (suspected) general effects to emphasize the importance and relevance of the topic.

Major point:

  • An English correction might be useful – sometimes misleading verbalizations are used.
  • Check that all abbreviations are explained in the manuscript (also abstract)
  • Could the authors please define the words “antibiotics”, “chemotherapeutic agents” and “antibacterial agents” or if they can be used interchangeable (….currently also called). To avoid confusion maybe the use of “antimicrobial chemotherapeutic agents” might be better.
  • Line 27-30: also the competition between different commensals should be mentioned besides the controlling effect of the immune system on colonization
  • In chapter 4: could the authors please emphasize the novelty and importance of their findings for the field.
  • Emphasize the development of resistances of sub-MICs also on other antimicrobials – this point is not so clear throughout the manuscript
  • Figure 1: please add a figure legend, which explains the used abbreviations and mechanisms.
  • Chapter 4.2: could the authors please explain the function of PSMs and the relevance of the pmt transporter (it is explained later in the chaper 5, however it is also needed here)?
  • And could you speculate how the increased pmt expression is involved in nisin A resistance – what is the proposed mechanism? (Line 255-256). Could you also explain the interaction between human-derived AMPs and Pmt / BraRS (as shown in Figure 2 and line 270/279)?
  • Conclusion: could you also mention the TCS as resistance mechanisms and not only your own work.

Minor point:  

Line 51-53: this sentence sound as if food preservations are administrated to cure infections.

Line 54-55: could you explain oac gens (proteins, genomic expression,…) and the function of ammonium compounds.

Line 61: not all AMPs and bacteriocidins are effective against MRSA. Better use “Some antibacterial peptides …”

Line 64-66: Please refer to the resistance mechanism and maybe to the effect of nisinA. It is explained in more detail later. However, for clearance it would be good to shortly mention this here and refer to the later chapter. Or delete this statement here.

Line 89: the role of lipid II has already been explained in line 81.

Line 100: TCS are predominantly found in prokaryotes (however, they are also found in archaea)

Line 111-113: the mentioning of PSMs as virulence factors regulated by agr would be useful here, since you refer to the pmt and PSMs later.

Line 118 please clarify that SaeRS is a TCS

Line 121: TSST-1; please don’t use unexplained abbreviations

Line 129: beta-defensins can also be produced by other cell types. But they are mainly produced by epithelial cells.

Line 130: here you use the name CAP18/LL37 later on you use LL37, please use always the same peptide name

Line 131: explain hBD3 and LL37

Line 132-135: could the authors please explain the mechanism of AMP repelling by the change in the surface charge in more detail or more clearly.  Also please refer to the nice Figure 1

Line 140: could you speculate on the mechanism of VraSR resistance to cellwall inhibitors.

Line 143: Are the other mentioned TCS not involved in bacteriocin resistance and only BraRS?

Line 185-191: Add a reference

Line 271: please modify the reference

Line 274: Hld is the gene name for the protein delta-toxin/delta-hemolysin. As the protein is transported, please refer to the protein name and not the gene name (please check this throughout the whole manuscript). PSMs are multiple peptides – please used the plural.

Line 275: PSMs are involved (plural)

Line 281-283: please add a reference to this speculation or speculate yourself

Line 295: how does an “unpredictable infectious disease” with a virus like SARS-CoV2 influence bacteriocidin resistance or are you referring to indirect AMP-resistance or antibiotic resistance? Please clarify your point.

Line 297: explain the correlation with clinically used antibiotics. It is not clear how these observations have an effect on antibiotics (VraRS…)

Reviewer 2 Report

It is a well written review article on the antibacterial peptides in S. aureus. It is timely too. One of the limitations of this review is that it focused only on MW2 strain. It could have reviewed the literature on USA300 strain as well.

Reviewer 3 Report

The review paper by Kawada-Matsuo et al. tries to give an overview on antibacterial peptide resistance in Staphylococcus aureus.  Some interesting observations and connections are made, but the paper reads more like an extension of the group's primary journal articles rather than an all encompassing review.  A review article can cite one's own work, but one does not do so as directly as what was written in this review.  We did this and our lab did that is not the way to present your critical findings in a review format.  The emphasis of this review was on your own work and narrowed to a nisin resistance overview.  A more expansive review is needed that includes the contributions of several other labs.  Several papers germane to this topic were left off, including Du et al., 2020; Jensen et al., 2020; Alves et al., 2020; Randall et al., 2018; and Barbosa     et al., 2021.

Other comments:

  1. Figure 2 was taken directly from other papers.  You should explain this in the Figure legend.
  2. Species names are always italicized and the specific epithet is all lowercase.
  3. Line 29  compromised hosts, such
  4. Line 54 The qac genes.  The genes themselves are not resistance factors, rather, they encode factors.
  5. Lines 64-66 Run-on sentence.
  6. Line 58  What are "these agents"?
  7. Line 100 Add a reference after bacteriocins
  8. Line 115 rot gene
  9. Lines 131-132 which possess a strong positive charge
  10.  Table 2 legend   compared to seven staphylococcal species.
  11. Line 181  S. aureus has several systems involved in resistance
  12. Line 263 BraRS mutations
  13. Line 264  highly resistant
  14. Line 287  antibacterial AMPs

Round 2

Reviewer 1 Report

In my opinion, all point have been addressed and the manuscript has been improved sufficiently. It is now a good review of the topic.

Reviewer 3 Report

The authors have addressed the concerns.